# Perceived Housing in Relation to Retirement and Relocation: A Qualitative Interview Study among Older Adults

**DOI:** 10.3390/ijerph192013314

**Published:** 2022-10-15

**Authors:** Erik Eriksson, Karla Wazinski, Anna Wanka, Maya Kylén, Frank Oswald, Björn Slaug, Susanne Iwarsson, Steven M. Schmidt

**Affiliations:** 1Department of Health Sciences, Lund University, 221 00 Lund, Sweden; 2Interdisciplinary Ageing Research, Faculty of Educational Sciences, Goethe University Frankfurt, 60323 Frankfurt am Main, Germany

**Keywords:** perceived housing, life course transitions, relocation, retirement, older adults

## Abstract

As people age the home environment becomes increasingly important. Retirement commonly leads to spending more time in one’s home, and relocating from your own home in older age could be associated with reduced health or wellbeing. The relationship between home and person is complex and perceived aspects of one’s housing such as social, emotional and cognitive ties are considered important factors for health and wellbeing. However, little is known about how perceived aspects of the home change in relation to retirement and relocation. This paper used Situational Analysis to explore, via situational mapping, how community dwelling older adults (aged 60–75) perceived their housing situation in relation to retirement and relocation. The results suggest complex relations between relocation/retirement and perceived housing, and between different aspects of perceived housing. Furthermore, the results suggest that the relationship between life transitions and perceived housing can be seen as bi-directional, where different life transitions affect aspects of perceived housing, and that perceived housing affects (decisions for) relocation. The results suggest complex relations between retirement and relocation, as well as other life transitions, and perceived aspects of one’s housing. It is important to consider these interactions to understand factors that affect health and wellbeing in older adults.

## 1. Introduction

As people age, a disparity between the older adult’s functional level and demands in the environment can affect the person’s ability to perform activities and restrict participation in everyday life. The home environment can also constitute a resource, supporting activities, routines, and participation [1,2,3,4]. The relationship between home and health is complex and affected by a multitude of factors, for instance by the physical environment, or by how a person perceives the environment and his or her capabilities in that environment. Social, emotional, and cognitive ties to a home are linked to wellbeing and identity, and the experience of one’s home as a supportive or hindering environment can change in connection to different life events. Even so, little is known about how the experience of the home and different life events interact and affect health and wellbeing in later life.

In this study we use the term *perceived housing* to cover different phenomena and concepts of experiences and symbolic representations related to living at home such as, but not exclusively, *meaning of home*, *housing-related control beliefs*, and *usability in the home* [5]. From a theoretical perspective, meaning develops over time and individuals experience home in a variety of domains leading to place attachment within a meaningful social and physical setting [6]. Having control over one’s environment is considered crucial for human development across the life course [7]. The concept of housing-related control beliefs was introduced to identify the role of proactive internal individual as well as more reactive external attitudes towards the home environment. The concept of usability derives from occupational therapy theories on person-environment activity transactions [8] as well as on the Ecological Theory of Ageing [9].

The importance of perceived housing for health and wellbeing in older age has been supported empirically. For instance, among very old people, aspects of perceived housing have been found to be associated with daily activities, wellbeing and depression [10,11,12]. Other findings indicate that these associations are also present in younger older adults around retirement age, i.e., 67–70 years old [13,14]. Although perceived housing is associated with both physical and mental health symptoms, independence in daily activities, and psychological wellbeing among different age groups in later life, we still know little about how these associations change across the life course, and when people experience different life transitions.

### 1.1. On Retirement and Relocation as Transitions Related to Perceived Housing in Later Life

Throughout our lives we experience events leading to individual changes that require some life adjustment. Such events, henceforth referred to as life transitions, or transitions, evolve from social and interpersonal transactions in ordinary social structures such as family constellation, occupation, residence, and social relationship. Even though there are transitions that are expected to be common for many people as we age, whatever constitutes a significant change in one’s life is likely to be a subjective assessment. Therefore, transitions are significant events that are variably experienced and are not necessarily perceived as critical. For some people, a transition runs smoothly without much effect on one’s life, while others might struggle with adjusting to the change. Others might even experience a transition as contributing as a positive change. In older age there are multiple life transitions that commonly occur, individually or together, of which retirement and changing housing, i.e., relocation, are arguably among the most common [15].

Transitioning into retirement can have varying effects on our lives. For example, retirement (with a pension) was found to have a protective association for depression when comparing to a working population [16]; however, the association differed between different contexts. Retirement has been found to be associated with higher depressive symptoms in the United States, and with lower depressive symptoms in Mexico and England [17], also suggesting contextual differences. There are older adults who experience ambivalence, anxiety, fear, and a feeling of loss, when they retire [18]. Retirement commonly leads to spending more time at home, which in turn can lead to changing experiences in the home. The home as such becomes an important setting for maintaining a sense of purpose after retirement [19].

Research has shown that the probability of relocating drops the later a person retires, and that marital status and physical functioning play significant roles in the decision to relocate [20]. Although not everyone wants to remain in their current home, and the reasons for relocation around retirement vary [21], most gerontological research has focused on ageing in place [22] or on relocation to residential care facilities [23,24], especially when the relocation is not planned or involuntary [25]. Due to the physical, psychological, and experiential separation from the home, relocation in such situations can be a challenging transition with adverse health outcomes [26]. Research that has investigated voluntary relocation among older adults, to congregate housing or assisted living, have shown that voluntary relocation to such housing options can be associated with beneficial effects on morbidity and mortality [27,28]. A review of the literature regarding factors influencing relocation for older adults to assisted living found that comprehending the need for relocation, and partaking in the decision to relocate, are factors associated with fewer negative outcomes (such as emotional distress, loss of self-worth, and inability to integrate into the new community), compared to forced relocation [29]. However, studies on the relationship between relocation within the ordinary housing stock and health and wellbeing for older adults are sparse. From a research perspective, studies have examined perceived housing and health in later life, but not explicitly from a transitional perspective. Despite growing research on the relation between perceived housing and health on the one hand, and life transitions and health on the other, there has been little effort to bring them together.

### 1.2. Aim

To the best of our knowledge, no studies have addressed perceived housing in relation to life transitions among people in the early stages of later life. In order to further comprehend influencing factors in the lives of older adults, it is important to study how life transitions that are experienced around earlier stages of later life interact with aspects of perceived housing.

The aim of the current article was to explore how different aspects of perceived housing relate to the experiences of relocation and retirement as two particular transitions in later life.

## 2. Materials and Methods

This qualitative study, inspired by Situational Analysis [30], is part of a larger project, Perceived Housing and Life Transitions: Good Ageing-in Place (HoT-Age). The project is a collaboration among researchers in Sweden and Germany.

### 2.1. Recruitment

Participants were recruited in Sweden and Germany. Eligible participants were 60 to 75 years old, lived in ordinary housing, had at least one life transition during the last 10 years, could participate in an interview for at least one and a half hours, and spoke either Swedish, German, or English. In Sweden, purposeful sampling was used to recruit a heterogeneous sample in regard to gender and age. In addition. In Germany theoretical sampling was used to recruit a variation in socioeconomic backgrounds, forms of housing (e.g., living alone or together with someone, dwelling), and different life course transitions. In Sweden, participants were recruited from the Stockholm and Skåne regions, via advertisement in monthly digital newsletters and on the websites of the two largest pensioner’s organisations. In Sweden, snowball recruitment was performed for the first 10 interviews and then discontinued due to high participation interest. In Germany, participants were recruited via the University of the Third Age in Frankfurt am Main, and via pensioner organizations across all Germany.

### 2.2. Participants

The data for the present study were collected during the SARS-CoV-2 pandemic. In total 50 people (34 in Sweden, 16 in Germany) in 49 interviews (in one interview there was one couple) were interviewed between 23 November 2020 and 4 May 2021. The current study included the 26 participants who had within approximately 10 years retired and relocated. Descriptive data were collected at the end of each interview. Of the 26 participants, 11 were women and 15 men, and ages ranged from 62 to 75 years old at the time of the interviews. See Table 1 for further characteristics.

Participants were categorized into two sequences depending on the chronological order of retirement and relocation. Cases that had relocated before retirement were denoted *sequence A*, and cases that had relocated after retirement were denoted *sequence B*.

### 2.3. Qualitative Interviews

The participants could choose whether they wanted to conduct the interview via telephone, video link (Zoom), or in person. Due to the SARS-CoV-2 pandemic, only two interviews were held in person, and the remainder were conducted via telephone (*n* = 4) or video link (*n* = 20). In Sweden, interviews were performed by authors EE or MK, and in Germany by author KW. All interviews were audio recorded.

A semi-structured interview guide, in which the participant was initially asked to freely talk about the events that were experienced as life transitions, and how those life transitions impacted her/his life, was used. Additional probing questions, how the life transition(s) affected perceived housing, daily functioning, social life, health and wellbeing, were asked in case the interviewee did not talk about such aspects of themself. The interview guide was piloted multiple times by several authors and amended through discussion within the research team. The interviews lasted between 48 min and 2 h and 21 min, with the mean of 1 h and 26 min. The interviews were transcribed verbatim by authors EE and KW, or by professional transcribers not affiliated with the research team. Participants were made anonymous during the transcription. The externally transcribed interviews were controlled for accuracy by randomly selecting and verifying four transcriptions.

### 2.4. Analysis

We developed situational maps inspired by Situational Analysis, which captures social phenomena in “situations”, understood as processual and emergent, meaning that they are continuously co-constituted, negotiated, and changed in interrelationships. Situational analysis involves mapping procedures, which visualize elements and relations in complex situations [30]. We adapted the mapping suggested by Clarke [30] to capture not only complex and relational situations, but also the process over time, which is especially relevant when studying transitions

During the first phase of analysis, initial and focused codes were created. The second phase consisted of developing situational maps. See Figure 1 for a flowchart of the analytical progression. For the Swedish cases, codes were translated into English during the creation of collective maps. For the German cases, translation into English was done during the initial coding. The translations into English were done by EE or KW, respectively. Communications among the co-authors during the analyses were in English.

The analyses started by creating initial codes through incident by incident coding using an inductive approach close to the material and the wording of the participants. Then, focused codes were created from the initial codes in an abductive manner based on themes that emerged from the material itself, as well as the theoretical framework of the project. The analysis was performed in NVivo (software ver. 20.3.0.535, QSR International Pty Ltd., Burlington, VT, USA) and Microsoft Excel (2016). Initial and focused coding was performed by authors EE and KW and agreed upon together with AW and MK.

We then used the focused codes to create multiple types and levels of situational maps, *individual maps* for each interview, *collective maps* for different aspects of perceived housing and sequences of transitions, and *joint maps* containing all aspects of perceived housing for sequence A and B respectively. The *individual maps* were visualizations of each interview, composed of a timeline of the experienced transitions and placing the focused codes in relation to respective transition. Microsoft PowerPoint (2016) was used to create the situational maps.

The focused codes from each individual map were merged into collective maps for each aspect of perceived housing, and each sequence of transitions (either sequence A or B). Consequently, two *collective maps* for each aspect of perceived housing were created. From the collective maps, macrocodes were created to describe groups of similar focused codes within each collective map. The macrocodes related to each aspect of perceived housing were placed in a *joint map* for each of the two sequences. Thus, the joint maps described macro-level codes of different aspects of perceived housing across the different transitions of retirement and relocation and enabled final analyses of the interrelationships between them.

To enhance trustworthiness [32], quality, and credibility [33,34], authors EE, KW, AW, MK, FO, and SS had multiple analysis meetings in which the codes and maps were discussed jointly. At a later stage in the analysis process, authors BS and SI contributed with critical reflections on the emerging results and the structure of the maps.

## 3. Results

The analyses resulted in two joint situational maps showing macrocodes for different aspects of perceived housing in relation to retirement and relocation. The aspects of perceived housing that stood out in the interviews were *Control Beliefs*, *Meaning*, *Usability*, *Practices*, *Social Relations*, *Future plans*, and *Neighbourhood*. The two joint maps are available as Appendix A and Appendix B. First, we present the descriptions of the results on basis of respective transitions, starting with experiences before both transitions, proceeding to experiences after retirement and relocation respectively. Thereafter we present our thoughts about housing related future plans. Within each descriptive section, similarities and differences between experiences of persons representing either Sequence A or B will be described.

### 3.1. Pre-Transitional Phase

In general, participants felt attached to their homes prior to experiencing retirement and relocation. However, how they perceived their home was not static—it had changed in participant’s lives with previous life course transitions, for example when children moved out, or when people separated from a partner:
*So between the two relationships, where I/There I made my own environment the way I wanted it, yes? So something like books on the floor or such. [laughs]*(German participant, Sequence B, Man, age 70)
*And at one stage when we were approaching retirement and the children had moved out, we lived in a townhouse which we felt was unnecessarily large, but the actual housing situation was very good*(Swedish participant, Sequence A, Man, age 74)

For those who relocated before retirement (Sequence A), the home was not only connected to family life and the respective transitions, but also, and to great extents, working life—for example, they would have studies or workplaces at home. In contrast, those who relocated after retirement (sequence B) described important places in the home as being connected to social life, e.g., recreation or living rooms, places enabling interaction and social relations.
*Yes, [at the old home] I had an ever larger workroom… Laughter...because there is was large areas, so it was an important workplace and for many years I had […] assignments in the school area, so I was chairman of the […] and had expedition at home and so on, so it was important to have such an environment.*(Swedish participant, Sequence A, Man, age 74)
*It was probably the recreation room, I think, because that was where we gathered, it was like someone could sit at the computer and someone could watch TV and that was where…. It was like where we lived, you could almost say.*(Swedish participant, Sequence B, Woman, age 62)

Accordingly, persons who relocated after retirement (sequence B) described social relations as an important factor of the home in older age, i.e., starting to worry of less socializing, or even becoming more isolated.

Beyond changes in the perception of home changing with past life events, participants also reasoned about the future and anticipated life events. This mainly manifested in accessibility issues, e.g., having several stairs in the entry, not having a lift in the building, and wanting a barrier free home. Thus, in general, most participants had already started thinking about how their current home was suitable for ageing even before they retired. These thoughts were, however, more pronounced in persons who relocated before retirement (Sequence A), who particularly worried about their ability to maintain their home in older age, a worry of becoming dependent on others in their own home. The fear of not being able to take care of the home, or oneself in the home, came from other events or transitions, such as functional decline due to illness, or the death of a partner.
*And then we started thinking, because we were just the two of us, who were left at home looking at each other between the floors, and then we had several floors that were not being used. And then [the husband] got a new knee and then I just realized that, no, we cannot age here, we cannot live here. We have to do something. And then you are faced with that situation, […], move, sell, or tear down and rebuild.*(Swedish participant, Sequence A, Woman, age 68)

These fears were largely absent in the narrations of persons who relocated after retirement (sequence B), and might have contributed to the earlier relocation of persons in sequence A.

### 3.2. After Retirement

The changes in perceived housing that participants described after retirement related to the meaning and practices of home, the social relations around the home, its usability, and the control beliefs related to the home. Again, we find many similarities in these narrated changes in perceptions of home after retirement, independent of whether participants relocated before or after retirement. Persons representing both sequences described how the new role and identity of being a retiree was linked to their housing practice, such as spending more time, and doing more activities, both at home and in the neighbourhood as well as fixing, changing, and maintaining the (old or new) home after retirement.
*I saw things that needed to be fixed at home. […] But… because we were in this situation that we would possibly move and so on, so for me was… I did a lot at home, on the property at home, but it still did not feel really meaningful, because I felt no, but this is… it’s probably doubtful that we will and stay here.*(Swedish participant, Sequence B, Man, age 68)

An experience of an increased sense of *control belief* would come from managing their responsibilities in and around the home—for instance, by renting the home or the car instead of owning they managed their situation to have fewer responsibilities. The increased sense of control belief could also come from an increased sense of autonomy and independence, by feelings of not having to follow societal norms anymore, being truer to oneself.
*[About retirement] That’s a different life. That’s a free life, no more dependency anymore.*(German Participant, Sequence A, Female, age 64)
*It is very nice as a pensioner not to have to do that. To not have to be on top all the time and try to look representative. So, because you do not have to wear the work uniform. Very nice.*(Swedish participant, Sequence A, Woman, age 68)

In line with the focus of their everyday lives shifting to the home and neighbourhood, their *social relations* also shifted to their (retired) partner, family, and grandchildren, and less to friends and neighbours. However, most participants also described how *neighbourhood* aspects was important to them, specifically how a variety of age in the neighbourhood felt invigorating, even though there was less socialization with neighbours.

Yet, despite many similarities, we could also identify differences in relation to perceptions of home depending on whether people relocated before or after retirement. Persons representing sequence A now described how refurbishments, or even renovations, focused on increasing *social relations* of the home, something that did not emerge as clearly before retirement. For instance, opening up the home by taking down a wall, or switching rooms to have larger rooms to socialize in, and thus enabling better social relations and interactions within the home. Persons representing sequence A described how important it was to have a more sense of space and openness both within the home and outside the home, aspects of *usability*, i.e., being able to view the life and nature outside the home by having large windows, or being able to see over the rooftops of other buildings.

Therefore, after retirement, aspects of *practice* had more focus on *social relations* for sequence A, aspects which sequence B already had in mind for future housing already before retirement.

Contrarily, persons in sequence B experienced how the sense of space changed when they got a new role as retiree, that they needed less space and felt that the current home was too big. This was similar to what was already experienced before retirement in connection to a change in parent role when their children moved out.

Turning to the anticipated future, we can again see that fears about becoming dependent on others and losing autonomy persisted also in the post-retirement stage. These fears were often initiated by other transitions such as the death of a partner or a reduction of functional capabilities due to illnesses, either one’s own or that of significant others (e.g., parents or a partner). However, some persons in sequence B also described this fear as the starting point of taking action in their decision-making process for a new home, a process that started before retirement already.
*But in connection to [mother] becoming wheelchair bound, we were just then rebuilding our bathroom and that made us very aware of this with accessibility. […] So for example then we did [a] shower that was easy to get into with a wheelchair. […] What also happened was that she was very young, she had not reached retirement age. […] So then I went through this of course that oh, it’s my mother, how will it affect me when I get older… and suddenly I got this insight, that oh… laughter… before I was like immortal and then I realized that oh, maybe I’m not immortal after all.*(Swedish participant, Sequence B, Woman, age 62)

### 3.3. After Relocation

After the relocation, descriptions from persons in both sequences showed they were being overall satisfied and happy with their new home and area, that they felt at home, and that the new home was more accessible and practical than the previous home. After relocation—and independent of whether this took place before or after retirement—these feelings of being at home were connected to aspects of suitability of the home and therefore of *usability*, specifically aspects of accessibility, spaciousness, or practicality. For instance, having a more spacious kitchen, a more open layout between rooms, or a great, open view outside, did not only increase feelings of being at home, but also made them perceive their new home as suitable for ageing. After relocation, persons in both sequences described increased perceived freedom and control (*control beliefs*). Perceived freedom was expressed in the experience of maintaining and creating their own home, letting go of burdensome maintenance, or connected to general relocation-related thoughts, for example to end an era and to start a new life, as one participant remembers.
*[B]ut [the move] was in some way liberating. […] Because then you could… I could end, you know, an era was over, if you say so.*(Swedish participant, Sequence B, Woman, age 70)
*The [old] house had, I’ll say about 260 square metres and they have to be taken care of. It took me over an hour just to water the flowers outside in the evening when [husband] wasn’t there, so from that point of view it’s a liberation.*(German participant, Sequence B, Woman, age 66)

After relocation, persons in both sequences described how they were renovating or refurbishing their new home to make it fit to this ‘era’ or life stage: For persons in sequence A there was descriptions of renovating the new home after relocation to improve the sense of connectedness to the outside world. For example, install more windows improving the view of life outside, aspects of openness, and therefore *usability*. In contrast, for persons in sequence B, renovating and refurbishing had more focus on *social relations*. For instance, by creating a sense of space and openness in the housing increased the possibility for socializing. In addition, in sequence B, there was also descriptions of refurbishing the new home to make it more accessible (*usability* aspects), for instance, by removing carpets and doors to make it wheelchair accessible.
*[W]e renovated quite a lot down here before we moved in, or in connection with us moving in. […] here in the living room where I sit today there were actually two rooms, so we tore down a wall and laid a new floor and a new ceiling, to get the space here in the living room. So that even when the grandchildren are a bit older, they will think it’s fun to visit grandma and grandpa”*(Swedish participant, Sequence B, Man, age 68)
*And as long as it is possible, as long as you have the opportunity to stay in your home, it is important that you set up your home so that you are satisfied, of course. […] And that you can receive guests and so on.*(Swedish participant, Sequence A, Woman, age 67)

There were mostly positive descriptions of *social relations* in the new home after relocation from persons in both sequences, and how the *neighbourhood* was important for activities and social relations after relocation. However, as previously described, social relations started to focus more (but not only) on family and grandchildren than on neighbours and friends. Some persons in sequence A described how the neighbourhood made them feel older, something that did not emerge in the narrative among persons in sequence B. For persons in sequence B, the importance of nature having nature close by emerged in their narrative after relocation, experiences that persons in sequence A described after retirement (see above).
*I have used the range of services [in the neighbourhood] to a much greater extent. […] Before, I drove around a bit more […] when you worked and you had to do things, so that… and now you have sort of found some of the needs here in the immediate area, so you feel that it is valuable that it is here and it feels good to take advantage of it.*(Swedish participant, Sequence A, Man, age 74)
*So, the problem if you say like this, with many older people, is that you move too late. That’s why I thought, we’re moving now while we’re still strong. Partly to be able to build the house, partly to be able to plan the house, partly and… It is that it takes time to get to know neighbours around. And neighbours are important, because they are part of the safety net. Now all of a sudden we have realized that we are the oldest on the street.*(Swedish participant, Sequence A, Woman, age 68)

For those who relocated before retirement, the relocation brought an increase in perceived housing autonomy—specifically that they felt like being able to decide by themselves, they could do whatever they wanted with the new home, and especially the sense that no one else decides over one’s life or home. This could manifest itself in reducing responsibilities or maintenance demand by having a rental apartment instead of owning it, relocating to a smaller place, or simply being happy to have relocated “in time”.
*But… it’s just that it’s quite nice to feel this, this is mine. […] I do not need to… ask anyone about anything, I do not need to do anything, I do not need to say anything […] so I decide for myself.*(Swedish participant, Sequence A, Man, age 68)
*Just that you want control over things in your life and take responsibility and I thought I did that when I moved so, I did not listen to anyone, but I ... do what I want. ….laugh…. […] Do you understand? It is control and responsibility.*(Swedish participant, Sequence A, Woman, age 73)

This was also the case for most persons relocating after retirement—hence representing sequence B. However, similar to the time before the retirement and relocation, some persons representing sequence B, described ambiguous feelings of not feeling completely at home, feelings connected to their social life. For instance, if they did not have good social relations in their neighbourhood, or if they needed to adapt and compromise with their partner’s routines and taste in interior design and objects.
*The flat was already completely furnished in a style that wasn’t mine, yes? So, what could you do? You could mix two styles. And I’m sometimes very hesitant about that. So I don’t push myself forward. That is, we kept the style and made a compromise.*(German participant, Sequence B, Man, age 70)

### 3.4. Post-Transitional Phase: Future Plans

Independent of whether participants relocated before or after retirement, they emphasized a wish to be able to remain in their (new) home in the future. Yet, this did not imply that their homes would stay exactly the same: Many participants planned to make changes to their new homes in order to have better accessibility, as well as being able to have better social relations. Whereas persons in sequence A wanted to refurbish their homes to improve *social relations*—especially to be able to accommodate guests—narrations of future renovations from persons in sequence B showed more focus on accessibility issues, i.e., *usability* aspects, to make the house more open, barrier free by removing obstructions such as steps, and better accessibility, i.e., installing handles, or renovating the bathroom.
*I would furnish the garage into a guest house [if I could change my home].*(Swedish participant, Sequence A, Woman, age 75)
*We live in […] a small one and a half storey house with living room, kitchen and bedroom on the ground floor and an upper floor with an extra bedroom and a small living room with TV, so you can see that in that case we cannot walk up the rather steep stairs to the upper floor, then we can live our life on the ground floor. […] Without making any major changes.*(Swedish participant, Sequence B, Man, age 68)

Most participants that relocated before retirement (sequence A) reported that they were now having a suitable housing for growing old in, even to the point that they would be able to remain in their new home for the rest of their lives. For some persons who relocated only after retirement (sequence B), however, the new housing did not completely fulfil their own criteria of a suitable housing and thus was not fully considered a home they could remain in for the future. Hence, persons who relocated only after retirement considered relocating again in the future. This was due to accessibility or spatial issues (*usability*) in the new home, not having good enough infrastructure in the new area (*neighbourhood*), or the fear of becoming dependent (*control beliefs*) and therefore wanting to have support by having children or relatives close by. The thoughts of relocating again in the future, was thus influenced by experiences of illness, i.e., reduction in functionality, leading to increased dependence, either own or that of the partner.
*And then… then you have to be open to having to move.*(Swedish participant, Sequence B, Woman, age 70)
*We should of course manage ourselves as long as possible, so it is not something we strive for, but when it no longer works, we have to imagine…*(Swedish participant, Sequence B, Man, age 75)

## 4. Discussion

This paper contributes to the understanding of perceived housing in relation to life transitions and health and wellbeing by exploring experiences of relocation and retirement in later life. As much of the existing relocation research has been conducted in the U.S., this study contributes with an understanding of events and life changes around retirement in two West-European countries—Sweden and Germany.

As expected, the results showed that relocation was often prompted by an accumulation of other events or transitions in the participants’ lives, such as the onset of a chronic illness or changes in family structure. The novel contribution to existing knowledge is that these transitions interacted with aspects of perceived housing, and thus affected how the participants experienced their home. That is, changes in perceived housing appeared to be a starting point for the process of considering relocation. In addition, the results suggest that the new role as retiree was related to spending more time at home as well as a change in social relations. These results are in line with previous research showing that the home becomes progressively more important after retirement, and that there is a shift to spending more time at home (e.g., [19,26]). Furthermore, changes in social relations could either be seen as a consequence of spending more time at home, or due to psychological changes as we age. For instance, the Socioemotional Selective theory [35] describes that as we age, tendencies are to reduce our social contacts and change our preferences for social interactions to partners or family, i.e., persons of more familiarity. The results showed that there were relations between different life transitions and perceived control of housing. In particular, the experience of reduced function due to a (chronic) illness led to changes in control beliefs, such as not being able to manage oneself or the home. Therefore, the analyses of the study indicated that the relations between life transitions and aspects of perceived housing were bi-directional, that life transitions affect perceptions of the home, and consequently, changes in perceived housing affect a (future) life transition. In this sense, perceived housing could be seen as mediating the relations between different life transitions, for instance, between a change in family structure and relocation. This suggestion would be in line with previous research indicating that children moving out affects choices of relocation [36,37].

In line with the previous literature [6], the results suggested associations between different aspects of perceived housing. In particular, changes in meaning of the home were closely linked to changes in the usability of the home. In addition, the results indicated a relation between housing-related control beliefs and usability; the perception of not being able to maintain the home was related to feelings that the home was too big, or that the home was not suitable for ageing in place. Even though changes in the dimensions of perceived housing were precursors to relocating, it was only when there were also changes in control beliefs that the relocation was carried into effect. For persons representing sequence A, this happened before retirement, and for persons representing sequence B, it happened after retirement. Worth noting is that for both sequences, there were changes both in meaning and control beliefs, leading to changes in usability before relocation. These findings underpin the need to consider how older adults think that they can deal with daily matters at home as an important dimension of perceived housing that may play a role in the decision to relocate in later life as well as having consequences for health and well-being. For example, there is much literature showing that general control beliefs are important for health and wellbeing as we age [38], and that high external housing-related control beliefs are associated with lower health outcomes, both among younger older adults [14] and among very old people [10]. However, our study does not shed light on whether changes in control beliefs as such, for instance without a simultaneous change in usability, could be sufficient for prompting relocation, which requires further investigation. An interesting observation is that for persons that relocated before retirement (Sequence A) there were little or no considerations of social relations for the new housing. In contrast, for persons who retired first and then relocated (Sequence B) there was, in addition to changes in other aspects of perceived housing, considerations of enabling social relations in the new home, considerations that would protect against future social isolations. Whether this was due to the sequencing of the life transitions remains unknown, and perhaps unlikely, because this difference in focus for the future home could be explained by a possible difference in civil status between persons in sequence A and B, where persons in sequence B reported, seemingly, a higher rate of being single.

The current study displays a complex picture of the many factors that influence relocation decisions in older age and longitudinal research is needed to understand how the different sequences operate over time taking different life transitions into account. In terms of the anticipation to move or stay around retirement age (i.e., 40- to 65-year-olds) a study conducted in the U.S. identified two major categories of relocating factors [39]. First, amenity factors (e.g., buying or ease of selling, taxes, availability of work as a source of income) and second, interpersonal and social influences of important others were found as significant push/pull factors. The authors concluded that both patterns are important and intertwined, but also that they seem to change the closer a person gets to a projected or planned retirement date. The results from the current study shed further light on the process of planning for future housing and the complex struggle of different ambivalent or counteracting aspects of perceived housing. Previous literature [19,40] has suggested that persons struggle with relocation, identity, and personal preferences, and that there are pronounced concerns about future housing arrangements or possible negative consequences from relocation due to being attached to the previous home. For instance, Kylén et al. [19] implies that societal norms influence where and how to live as we age, and societal norms generate ambivalent feelings of identity and relocation decisions. From our results, it could be suggested that this was the opposite experience, where persons rid themselves of societal norms, feeling free and true to themselves. The riddance of societal norms, and more specifically the freedom and autonomy related to that action, might have been an important factor for successful identification and attachment with a new home, and hence having a successful relocation would warrant further investigation. The results from this study further suggests that whether a person relocated before or after retirement was not reflected in systematic differences in perceived housing. That is, the same relations, either between life transitions and aspects of perceived housing or between different aspects of perceived housing, appeared in the analyses regardless of sequence of events.

### 4.1. Strengths & Limitations

This study was exploratory, an approach to research that is beneficial when the topic has not yet been thoroughly investigated. Striving for diversity in experiences of perceived housing, retirement, and relocation our recruitment strategy involved making use of multiple pensioner’s organizations in different regions in the two countries. The study participants ranged from 62 to 75 years old and were heterogeneous in regard to sex and type of housing. However, the recruitment method might have attracted mostly proactive and resourceful persons (e.g., high education level), already engaged in matters considered relevant by older adults, such as housing issues. For that reason, there is a risk that the study missed the experiences and viewpoints of older adults who are not active members of senior citizen organizations. Because we only were able to control for a balanced sample in regard to age and sex, the recruitment did not specifically consider certain transitions or specific sequences thereof, nor certain experiences of perceived housing. This is specifically problematic when multiple phenomena are part of the study. Thus, although we gained rich and varied data from the participants’ narratives, we do not claim that saturation was achieved.

Turning to the setting of data collection, due to the COVID-19 pandemic only five of the interviews were held in person. Prior to the study, we assumed that doing remote interviews using digital technology (i.e., video link) would be a limitation. While this does apply to some extent, especially in terms of the possibilities for the participants to show the interviewer his/her home, collecting data online went much better than anticipated. In addition, many participants had used online communication platforms before, which exemplifies that the participants represent a digitally resourceful segment of the population of older adults.

### 4.2. Implications for Further Research

In order to further explore our understanding of how relocation, retirement, and the experience of the home environment interacts in later life, it would be imperative to recruit additional diverging experiences of retirement and relocation. In addition, the mediating or possibly moderating roles of different aspects of perceived housing to different life transitions should be studied in order to better understand the underlying processes and experiences that constitute the basis for relocation decisions.

## 5. Conclusions

The results from this study suggest that there are complex relations between life transitions and perceived housing in an early phase of later life, and between different aspects of perceived housing. In particular, the results showed how different life transitions, such as changes in family structure or the onset of a chronic illness leading to reduced function, related to changes in aspects of perceived housing, in particular aspects of meaning and control beliefs. Furthermore, the results showed how aspects of meaning or housing-related control beliefs were associated with aspects of housing-related usability. Furthermore, the results showed how the relations between these aspects of perceived housing contributed to thoughts of relocating, and consequently making the decision to relocate. Accordingly, the results suggest that the relationship between life transitions and perceived housing can be seen as bi-directional. That is, life transitions affect aspects of perceived housing, and aspects of perceived housing affect (decisions for) future life transitions.

## Figures and Tables

**Figure 1 ijerph-19-13314-f001:**
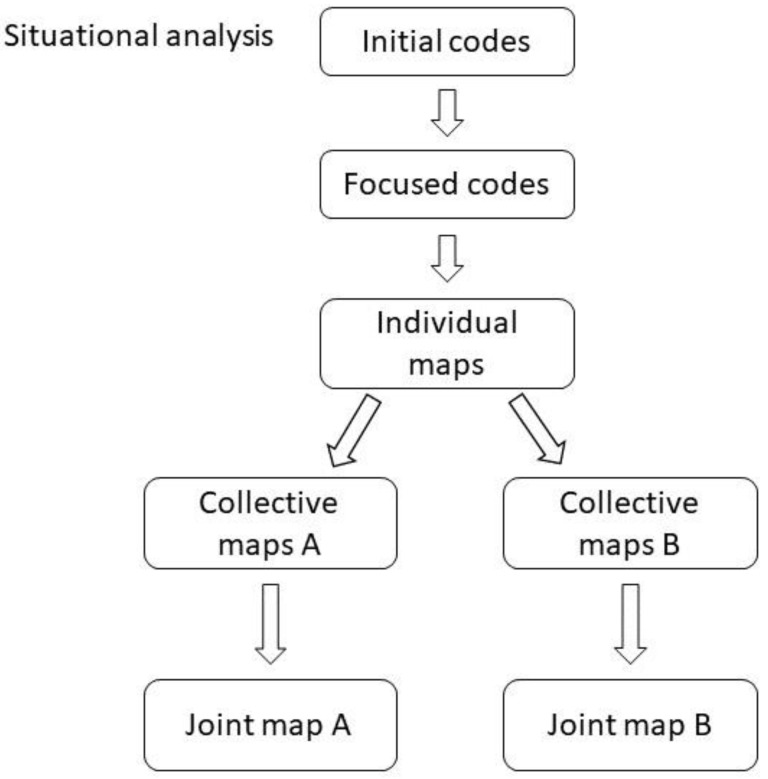
Flowchart describing the analytical progression inspired by Clarke [30]. A represents Sequence A, i.e., relocation happened before retirement, and B represents Sequence B, i.e., retirement happened before relocation.

**Table 1 ijerph-19-13314-t001:** Description of the participants (N = 26).

Characteristic		Sequence A	Sequence B
		n = 16	n = 10
Age	Median (IQR)	68.5 (65–74)	70 (66.5–75)
	Range (Min–Max)	10 (65–75)	13 (62–75)
		*n* (%)	*n* (%)
Sex	Female	7 (43.8)	4 (40.0)
	Male	9 (56.3)	6 (60.0)
		*n* (%)	*n* (%)
Native born	Yes	13 (81.3)	8 (80.0)
	No	3 (18.8)	2 (20.0)
		*n* (%)	*n* (%)
Education	Primary school (9 years)	0 (0.0)	0 (0.0)
	Upper secondary school	2 (12.5)	2 (20.0)
	Vocational	1 (6.3)	1 (10.0)
	University	10 (62.5)	6 (60.0)
	Doctoral degree	3 (18.8)	1 (10.0)
		*n* (%)	*n* (%)
Civil status	Relationship—Living together	12 (75.0)	6 (60.0)
	Relationship—Living apart	2 (12.5)	0 (0.0)
	Single	2 (12.5)	4 (40.0)
		*n* (%)	*n* (%)
Housing type	Apartment	8 (50.0)	6 (60.0) *
	House	8 (50.0)	4 (40.0) *
	Camper	0 (0.0)	1 (10.0)
		*n* (%)	*n* (%)
Self-Rated Health ^a^	Excellent	5 (31.3)	2 (20.0)
	Very good	5 (31.3)	5 (50.0)
	Good	5 (31.3)	2 (20.0)
	Fair	1 (6.3)	1 (10.0)
	Poor	0 (0.0)	0 (0.0)
Retirement age	Median (IQR)	65 (63–67)	63 (61–65)

IQR = Interquartile range. * One participant replied both Apartment and House. ^a^ SF-12 [31].

## Data Availability

Data cannot be shared publicly because it contains sensitive information about the participants, and they did not provide consent for public data sharing. Codes and maps that emerged through the analysis can be shared upon request.

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
