# Peer review of "Perceived Housing in Relation to Retirement and Relocation: A Qualitative Interview Study among Older Adults"

_ijerph, 2022, doi:10.3390/ijerph192013314_

Round 1

Reviewer 1 Report

Article prepared carefully. A good introduction to the subject. Interesting idea for research. The article contains interesting conclusions corresponding to the assumed goals. The summary in the form of maps describing the strengths and weaknesses of the current life situation of seniors is well illustrated. The weakest point of the article is the small sample size. I am aware that this amount is not subject to changes in this version of the publication. In this regard, I leave the final decision to the editors.

Author Response

Thank you for this important comment. Even though there is no set rule for sample size in qualitative inquiries, we feel that the comment merits considerations and discussion.

What constitutes an “enough” size for qualitative research depends on multiple factors. For instance, the purpose of the inquiry, what is at stake, what would be useful, or what will have credibility. Smaller samples can, by an in- depth design, provide significant understanding of the phenomena under investigation. As we had an exploratory approach, with in-depth interviews recruited either via purposeful or theoretical sampling, 26 participants could arguably be an enough sample for analyzing the phenomena under investigation.

However, whether we reached saturation, i.e. whether we can reasonably say that we have covered most experiences of those phenomena in that population, is something we cannot claim, which is also discussed in the article (lines 556-570).

Reviewer 2 Report

This manuscript has an interesting research topic and a clear purpose. I very much enjoyed reading this research. Please consider a couple of comments.

1.       This research aims to understand perceived housing in relation to retirement and relocation. The introduction was well-organized and well-written. The introduction highlighted the significance of this research because there is no research considering perceived housing, health, and life transitions together (page 2 and 3). However, the research findings and discussion sections failed to provide a strong argument for this perspective. That is, research findings described different aspects of perceived housing related to the experience of relocation and retirement, but failed to provide enough information regarding the health and well-being of research participants. As the authors described in conclusion, this research suggested that there are complex relations between life transitions and perceived housing in an early phase of later life and between different aspects of perceived housing. Thus, there is a discrepancy between the purpose of this research and its significance of this research. Also, two appendix diagrams did not describe issues related to health and well-being. Thus, this research needs more strong argument about its significance of this research.

2.       Line 144 on page 5 of 17, “See table 1” is inappropriate in this sentence.

3.       Line 164, I think the citation should be added after by Clarke.

4.       For the recruitment, please add a more specific reason why this research interviewed with the research participants being 60 to 75 years old. Participants were recruited in Sweden and Germany, but the sampling processes were different in the two countries. Is there any specific reason?

5.       Table 1 – Sequence B (Housing type) indicated 11 even though the total population was 10. Please double-check the numbers of the table 1 (housing type).

Author Response

  1. We appreciate this valuable comment as it was not our intention to relate aspects of perceived housing, relocation, and retirement to health and wellbeing in the current study. We have revised the text before and in the aim to remove further confusion. Hopefully the readers will find the aim of the study more clear now.
  2. Thank you for noticing this inaccuracy in the text, it has now been removed.
  3. We agree with the comment, which now have been amended accordingly
  4. Thank you for these comments and for giving us an opportunity to elaborate on this.

    As we wanted to capture fairly recent experiences of retirement of young-old persons, it was important to recruit with an age-range around retirement age. Retirement can happen at different ages for different reasons. Therefore, to allow for retirement at different ages, and to allow a relocation to have happened somewhat close in time to the retirement, we decided to recruit for an age range of 60-75. The actual age range of the interviewed persons were 62-75.

    Regarding the different sampling processes in Sweden and Germany.

    The reason was due to the ethical application/approval, which differed in the countries. In Germany, the ethical approval allowed for questioning potential participants about experiences before the interview, which then allowed for theoretical sampling process. In Sweden it was not possible to ask for such information from the participants before inclusion/interview due to the ethical application/approval, which is why, in Sweden, the sampling process was to purposefully diversify inclusion to age and sex.
  5. The discrepancy between responses (11) and responders (10) is due to one participant replying both House and Apartment. We have improved the description of the asterisk (*) under Table 1 to better reflect this discrepancy. Hopefully, it is now more clear for the reader.